# Sexually transmitted infections and factors associated with risky sexual practices among female sex workers: A cross sectional study in a large Andean city

Luz Marina Llangarí-Arizo[1,2,3]*, S. Tariq Sadiq[4,5], Cynthia Márquez[1], Philip Cooper[1,5], Martina Furegato[4], Liqing Zhou[4,5‡], Luisa Aranha[6‡], Miguel Martín Mateo[1,2,3‡], Natalia Romero-Sandoval[1,3‡]

1 School of Medicine, Universidad Internacional de Ecuador, Quito, Ecuador, 2 Unitat de Bioestadística, Facultat de Medicina, Universitat Autónoma de Barcelona, Barcelona, Spain, 3 Red Internacional Grups de Recerca d'Amèrica i Àfrica Llatines – GRAAL, Barcelona, Spain, 4 Applied Diagnostic Research and Evaluation Unit, St George's University of London, London, United Kingdom, 5 Institute for Infection & Immunity, St George's University of London, London, United Kingdom, 6 Ministerio de Salud Pública de Ecuador, Quito, Ecuador

☯ These authors contributed equally to this work.
‡ These authors also contributed equally to this work.
* lullangari@uide.edu.ec

**Data Availability Statement:** All relevant data are within the paper and its Supporting information files.

## Abstract

### Background

There are limited published data on factors related to risky sexual practices (RSP) affecting sexually transmitted infections (STIs) among female sex workers (FSWs) in Ecuador.

### Methods

Cross-sectional study of FSWs presenting for a consultation in a primary health care centre during 2017. A questionnaire was administered to collect information on RSP and potential risk factors including age, membership of an FSW association, self-report of previous STI diagnosis, previous treatment for suspected STI and temporary migration for sex work. Associations between RSP and potential risk factors were estimated by logistic regression. The proportion of STI was estimated from vaginal swabs by real-time PCR for four sexually transmitted pathogens (*Neisseria gonorrhoeae*, *Trichomonas vaginalis*, *Chlamydia trachomatis*, and *Mycoplasma genitalium*).

### Results

Of 249 FSWs recruited, 22.5% had reported RSPs at least once during sex work. Among FSWs reporting unprotected vaginal sex in the previous three months, 25.5% had at least one other RSP type. 17.6% (95%CI 13.3–22.8) had at least one active STI. Prevalence of co-infections was 2.4% (95%CI 1.1–5.2). In multivariable analysis, RSP was associated with age (adjusted OR 1.06; 95%CI 1.02–1.10), membership of an FSWs association (aOR 3.51; 95%CI 1.60–7.72) and self-reported previous STI (aOR 3.43; 95%CI 1.28–9.17).

**Funding:** This study was supported by a grant from the University Internacional del Ecuador [Grant Number UIDE-EDM04-2016-2017] https://www.uide.edu.ec/, Wellcome Trust Institutional Strategic Support Fund https://wellcome.org/what-we-do/our-work/institutional-strategic-support-fund and St George's, University of London https://www.sgul.ac.uk/. The funders had no role in the study design, data collection, and analysis, decision to publish, or preparation of the manuscript.

**Competing interests:** The authors have declared that no competing interests exist.

## Conclusions

Among a population of female sex workers with high proportion of STIs, increasing age and belonging to an FSWs association was associated with a higher likelihood of engaging in RSP with clients. Engaging with FSWs organisations may reduce the burden of STI among sex workers.

## Introduction

An estimated 376 million new infections worldwide occur annually with one of four sexually transmitted infections (STIs): chlamydia, gonorrhoea, syphilis and trichomoniasis [1]. In addition, for gonorrhoea, there has been a significant increase in global prevalence of antimicrobial resistance (AMR), particularly to fluoroquinolones and azithromycin as well as worrying emergence of AMR to extended spectrum cephalosporins [2]. Importantly, less is currently known about AMR in South America.

Risky sexual practices (RSP) are defined as any sexual activity that increases the risk of contracting STIs [3]. This may involve sexual activity with multiple partners, inconsistent condom use, having sex under the influence of drugs, or initiating sexual activity before the age of 18 years [4]. Female sex workers (FSWs), due to the dynamic of sex work and other individual and social circumstances, face several factors that make them vulnerable to RSP and therefore to STIs. Previous studies show that age, migration for sex work, place of soliciting clients or place where FSW engage in sex and community organization are associated with RSP [5–8]. On the other hand, some studies mention that a prior history of STI could be an indicator of RSP [9, 10]. In Ecuador, previously published data on RSP and STIs risk have focused largely on HIV infection [11, 12], men who have sex with men [13, 14] or transgender persons [15] while information for female sex workers (FSWs) is outdated and limited [16, 17]. In developing countries there are few health programs in sex work environments coupled with limited STI surveillance and weak or non-existent structural approaches to RSP [18, 19].

Sex work is legal in Ecuador where it operates either outdoors or within closed establishments (known as "*casas de tolerancia*" [20, 21] or brothels. All brothels must be licensed, and FSWs, to work within such brothels, must obtain an "Integral Health Carnet" (IHC), an occupational license provided by the Ministry of Health that registers medical care, screening tests and vaccines, as well as educational and prevention activities, that is valid for 1 month. To obtain and maintain a validated IHC, FSWs must be over 18 years and regularly test negative for syphilis and HIV [22].

FSWs' occupational associations aim to support occupational rights and to protect against violence and police harassment mainly when the work is on the street [21]. Sex work is present throughout the country but is concentrated in large cities where it is a major social and health issue related to poverty [20].

Exploring factors related to RSP is important because this information can be biased or limited due to the stigma and discrimination that exists around sex work: better knowledge of RSP among FSWs helps to improve the effectiveness of prevention programs.

In the present study, we analysed potential factors associated with RSP among FSWs in the Andean city of Quito. The specific objectives of this study were to identify factors related to unprotected oral, vaginal, or anal intercourse and to estimate the proportion of common STIs (gonorrhoea, trichomonas, chlamydia, and *Mycoplasma genitalium*) in a population of FSWs.

These pathogens were considered together because of shared symptomatology and to provide a broader view of the potential public health consequences of RSP.

## Materials and methods

### Study design and sample collection

A descriptive cross-sectional study was conducted among FSWs over 18 years of age in a primary health centre in Quito, a large Andean City, during the last quarter of 2017. The women were enrolled using a convenience sample.

Women attending for a health check-up at a public clinic required for validation of their IHC to allow them to continue to engage in sex work. In addition, women members of an association of FSWs were contacted through visits to their workplaces (streets and "casas de tolerancia") and asked to attend the same clinic for evaluation. FSWs where excluded if they attended the medical consultation more than once, if pregnant or if unwilling to provide questionnaire data. All participants provided written informed consent and none of them received any compensation for their participation. On each sample collection day all FSWs attending the clinic and willing to participate were enrolled consecutively prior to their medical consultation. A single study identifier was used to link questionnaire with the biological samples.

The number of female sex workers who agreed to participate, 249, makes it possible to estimate an expected prevalence of, 20% + -5%.

Participants were interviewed by a local study investigator (LMLLA) using a questionnaire designed for the study, provided as a S2 File.

### Variables

Included variables were age, membership of an FSW association, whether participants had temporarily left the city for sex work, self-report of being previously diagnosed with an STI (gonorrhoea, trichomonas, chlamydia, syphilis) and previous treatment (oral antibiotics, 'pain pills', vaginal ovules and creams) for suspected STI or genital infection at some point during sex work prescribed by a physician, having previously chronic pelvic pain at least once during their sex work, and number of medical and health-related consultations in the previous year.

The variable of interest in this study is RSP that was defined as having oral (UPOS), vaginal (UPVS), or anal (UPAS) intercourse without a condom at some point during sex work and during last three months.

### Clinical samples collection and DNA preparation

A vulvo-vaginal sample was collected using Xpert® CT/NG patient-collected Vaginal Swab Specimen Collection kit (Cepheid), in addition to that required for routine clinical evaluation. DNA was extracted using the PureLink Genomic DNA Mini Kit (Thermo Fisher Scientific) according to manufacturer´s instructions. The quantity and quality of the DNA were measured by spectrophotometry, using the Nanodrop ND-1000 (Thermo Scientific).

### PCR screening for STI

Real-time polymerase chain reaction (RT-PCR) was done for *N. gonorrhoeae*, *T. vaginalis*, *C. trachomatis* and *M. genitalium*. Gene targets and primers for PCR detection are shown in S1 Table. RT-PCR was done using Applied Biosystems 7500 Fast Real Time PCR System in a volume of 10 μl/reaction containing 5μl of Taqman™ Fast Universal PCR Master Mix, 1μl of 10x Exogenous Internal Positive Control (IPC) Mix, 0.2 ul 50x IPC DNA, 250 nM each of the

primers, 100 Nm each of the probes and 50 ng of templates DNA. The cycling parameters were 95˚C for 10 min, followed by 40 cycles of 95˚C for 15 s. 60˚C for 1 min.

Currently diagnosed with an STI was defined as having at least one positive reaction for any of the 4 pathogens and co-infection was defined as detection of two or more pathogens. Four samples were insufficient for PCR and these individuals were excluded from the estimation of the proportion.

## Statistical analysis

Data entry and analysis were done using IBM® SPSS® version 24 computer software. Comparisons of age by FSW membership of an association and RSP was done using t-tests. A p-value < 0.05 was considered statistically significant. Univariate and multivariable logistic regression models were used to explore associations between RSP and potential factors.

## Ethical considerations

The study was conducted according to the Declaration of Helsinki and approved by the Ethical Committee of Universidad Internacional del Ecuador (02-02-17).

## Results

### Demographic and social characteristics

Of 251 FSWs invited, 249 consented to participate. Median age of participants was 35 years (range 18–61 years). Among participants, mean age was higher in those that belonged to a FSW association (78/249; 31.3%) compared to those not belonging (40.1 years vs. 32.9, p<0.001) but was similar among those reporting that they temporarily migrate for sex work (79/249; 31.7%) compared to those who did not (34.1 years vs 35.6; p> 0.05). 24 (9.6%) self-reported a previous STI diagnosis at least once during their sex work (Table 1); previous gonorrhoea was reported by 13(5.3%), trichomoniasis by 6 (2.4%), chlamydia by 5 (2.0%) and syphilis by 6 (2.4%) participants. About twenty five percent of respondents had previous chronic pelvic pain. 185 (74.3%) declared treatment for an STI at least once during their sex work. 34 (13.7%) had only one medical visit in the last year including this study visit while approximately 50% reported between nine and eleven consultations which they received at least one of the following health-related services: information about STIs; examination or treatment for STIs symptoms; HIV test and prevention information, and receipt of free condoms and instructions on their use.

### Risky sexual practices (RSP)

Fifty-six (22.5%; 95% CI 17.8–28.6) FSWs reported engaging in RSP at least once during sex work. Among the 196 women who declared protected vaginal sex, three had some RSP: 1/196

**Table 1. Background characteristics of female sex workers, Quito, 2017.**

| Background Characteristics | % or Mean | Number |
|---|---|---|
| Age (in years) | 35.2 | |
| Membership of an FSWs association | 31.3 | 78 |
| Self-report of a previous STI diagnosis | 9.6 | 24 |
| Currently diagnosed with an STI | 17.3 | 43 |
| Previous treatment for suspected STI | 74.3 | 185 |
| Temporary migration for sex work | 31.7 | 79 |
| Risk sexual practices (RSP) | 22.5 | 56 |

(0.5%) UPAS and 3 (1.5%) UPOS. The mean age for those in the RSP group was greater than those in the non-RSP group (40.8 vs. 33.5 years, p<0.001).

In the last three months, 47/53 (88.7%) women who reported having vaginal sex had UPVS, 7/8 (87.5%) women who reported anal sex had UPAS and 12/12 (100%) women who reported oral sex had UPOS. Of the FSWs that reported UPVS in the last three months, 12/47 (25.5%) had at least one other RSP: 7/47 (14.9%) UPAS and 9/47 (19.1%) UPOS.

### Vulvo-vaginal proportion of STI

Of the 245 vulvo-vaginal samples, positivity rates were: *N. gonorrhoeae* (NG) 1.2% (95%CI 0.4–3.5), *T. vaginalis* (TV) 9.8% (95%CI 6.7–14.2), *C. trachomatis* (CT) 4.9% (95%CI 2.8–8.4), and *M. genitalium* (MG) 4.9% (95%CI 2.8–8.4). Overall, 17.6% (95%CI 13.3–22.8) of FSWs had at least one STI. Co-infection was detected in 2.4% (95%CI 1.1–5.2) and they were: 2 CT/TV, CT/NG, NG/TV, CT/NG/TV, and CT/TV/MG. Of all women currently infected, four (9.3%) reported having had any STI in the past.

### Factors associated with RSP

Univariate analyses factors showed significant association between RSP with age, membership of an FSWs association, self-report of a previous STI diagnosis and currently diagnosed with an STI. Previous treatment for suspected STI and temporary migration for sex work were not significant. In multivariable analysis showed that age (adjusted OR 1.06, 95%CI 1.02–1.10), membership of an FSW association (aOR 3.51; 95% CI 1.60–7.72) and self-report of a previous STI diagnosis (aOR 3.43 95% CI 1.28–9.17) remained significant Table 2.

## Discussion

In the present study, we set out to improve our understanding of risky sexual practices (RSP), understood to mean genital or extragenital sexual intercourse without a condom, among a group of FSWs working in poor areas in central Quito, a large Andean city. Almost one in four FSWs participating in this study reported RSP, and among the FSWs that reported unprotected vaginal sex in the last three months, 25.5% had at least one other RSP type.

This study reported that RSP was independently and positively associated with age, membership of an FSW association and previous STI diagnosis. Risk by age increased 1.06 odds for each year and is consistent with previous studies reporting older FSWs to be more likely to engage in RSP to attract clients [5, 23].

Previous studies have reported that FSWs belonging to an association, support networks or peer groups are better at negotiating use of condoms with clients [24]; however, the results of our study showed that belonging to an FSWs' association increased the likelihood of engaging in RSP independently of age. This could be explained by the fact that FSWs belonging to the association are more likely to be engaged in informal sex work on the street and consequently

**Table 2. Univariate and multivariable analysis of factors associated with RSP among FSWs of this study.**

| Variables | n | (%) | Crude OR | 95% CI | p- value | Adjusted OR | 95% CI | p- value |
|---|---|---|---|---|---|---|---|---|
| Age | Mean, 35.2 years | | 1.09 | 1.05–1.13 | <0.01 | 1.06 | 1.02–1.10 | 0.002 |
| Membership of an FSWs association | 78 | 31.3 | 4.26 | 2.28–7.94 | <0.01 | 3.51 | 1.60–7.72 | 0.002 |
| Self-report of a previous STI diagnosis | 24 | 9,6 | 5.00 | 2.10–11.92 | <0.01 | 3.43 | 1.28–9.17 | 0.01 |
| Currently diagnosed with an STI | 43 | 17.3 | 2.16 | 1.06–4.40 | 0.03 | 1.29 | 0.57–2.94 | 0.54 |
| Previous treatment for suspected STI | 185 | 74.3 | 0.83 | 0.42–1.61 | 0.577 | 1.25 | 0.56–2.78 | 0.59 |
| Temporary migration for sex work | 79 | 31.7 | 1.09 | 0.57–2.06 | 0.802 | 1.80 | 0.83–3.93 | 0.14 |

have a reduced capacity to negotiate condom use [21, 25] to attract clients. Enforcing condom use is probably easier in the more 'regulated' and protected environment of brothels [8]. In this study, the main role of the FSWs' association was to protect unregulated working spaces in cities [20, 21] and no to ensure implementation of effective preventive health education messages. Therefore, it is important to focus on these particularities of sex worker groups to improve STI prevention programs.

In this study, temporary migrations between cities, that serve to obtain more clients and income, were not affected by age, and did not affect RSP. Other studies show that some mobile FSWs groups are unable to turn away clients for unprotected sex [5, 6].

This study observed that a self-report of a previous STI diagnosis was associated with RSP but a previous STI treatment was not. The discrepancies may be due to recall bias or social desirability bias related to self-reporting of STIs. Nevertheless, information about STI diagnosis could be an indirect indicator of unprotected sexual behaviour [9].

The inconsistent condom uses in FSWs increase the risk of contracting STI through multiple factors including a large number of sexual partners, unsafe working conditions, and barriers to negotiation a consistent condom use [26]. Moreover, their vulnerability is exacerbated by the social and health inequalities which they face [27, 28].

In this study, the proportion of *N. gonorroheae* (1.2%) and *C. trachomatis* (4.9%) was lower than the reported in previous studies in Mexico (2.9% and 15.3%) [29] and Perú (1.6% and 16.4%) [30]. Regarding, the proportion of *T. vaginalis* (9.8%), it was higher than the rates reported in Peru (7.9%) [30], China (2.1%) [31] and Iran (6.1%) [32]. We observed a proportion of *M. genitalium* of 4.9%, somewhat lower than a 15.9% estimated among FSWs worldwide from a meta-analysis of studies between 1991 and 2016 [33]. To our knowledge, this is the first estimate of STI proportion in a group of FSWs in Ecuador using a highly sensitive molecular method (RT-PCR). Because of the sampling method, our study findings are not necessarily representative of the entire population of FSWs. The low relatively proportion of STIs observed here could indicate the importance of regulating and legalizing sex work and allowing FSWs access to health services. However, more data are needed to understand better the different elements of sex work at both city and country levels.

The study had several potential limitations. Because all participants were FSWs attending a state health clinic for validation of their IHC, there may have been reduced heterogeneity with respect to potential risk factors. We did not study various risk factors that might have underestimated RSP such as sexual behaviour with stable or occasional non-paying partners, educational level, and alcohol or drug use [23]. The study recruited FSWs attending health clinics for IHC validation potentially leading in bias resulting from pressure to provide 'correct' responses [34]. There was also potential selection bias as a significant number of FSWs, not represented in this sample, are thought to engage informally in RSP without IHCs. Finally, the findings of this study may not be generalizable to FSWs working outside Quito in smaller cities and towns, where FSWs tend to be less organized and more vulnerable and consequently may have higher rates of RSPs.

## Conclusions

In Quito, older FSWs, previously diagnosed with an STI and belonging to an FSW association, are more likely to engage in RSP with their clients. This study provides new information about the current proportion of STIs in FSW. Our data provide surprising and important elements that will inform and strengthen the capacity of public health interventions to reduce RSP and impact of STI among FSWs, key factors to improve their sexual health.

## Supporting information

**S1 Table. Gene targets and primers used for current diagnosed of STIs in this study.**
(DOCX)

**S1 Data.**
(XLSX)

**S1 File. Regression model.**
(PDF)

**S2 File. Questionnaire English and Spanish version.**
(PDF)

## Acknowledgments

The authors would like to thank the FSWs associations for supporting the development of this study.

## Author Contributions

**Conceptualization:** S. Tariq Sadiq, Cynthia Márquez, Philip Cooper, Natalia Romero-Sandoval.

**Data curation:** Luz Marina Llangarí-Arizo.

**Formal analysis:** Luz Marina Llangarí-Arizo, Martina Furegato, Miguel Martín Mateo.

**Funding acquisition:** S. Tariq Sadiq.

**Investigation:** Luz Marina Llangarí-Arizo, Cynthia Márquez, Martina Furegato, Liqing Zhou, Luisa Aranha, Miguel Martín Mateo.

**Methodology:** S. Tariq Sadiq, Liqing Zhou, Luisa Aranha.

**Project administration:** Cynthia Márquez.

**Resources:** Philip Cooper, Luisa Aranha.

**Software:** Martina Furegato.

**Supervision:** S. Tariq Sadiq, Cynthia Márquez, Philip Cooper, Miguel Martín Mateo, Natalia Romero-Sandoval.

**Validation:** Philip Cooper, Liqing Zhou, Miguel Martín Mateo.

**Writing – original draft:** Luz Marina Llangarí-Arizo.

**Writing – review & editing:** S. Tariq Sadiq, Cynthia Márquez, Philip Cooper, Martina Furegato, Liqing Zhou, Luisa Aranha, Miguel Martín Mateo, Natalia Romero-Sandoval.

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
