## [Decision Letter · Decision Letter 0]

23 Dec 2020

PONE-D-20-32212

Sexually transmitted infections and behavioural factors associated with risky sexual practices among female sex workers: a cross sectional study in a large Andean city

PLOS ONE

Dear Dr. Llangarí,

Thank you for submitting your manuscript to PLOS ONE. After careful consideration, we feel that it has merit but does not fully meet PLOS ONE’s publication criteria as it currently stands. Therefore, we invite you to submit a revised version of the manuscript that addresses the points raised during the review process.

You need to address the methodological and statistical issues pointed out by the referees. There is also the question of the alignment of the conclusions to the aims and hypotheses in the paper as currently these are not in agreement. I suspect that some of these issues might be a question of translation.

We look forward to receiving your revised manuscript.

Kind regards,

Andrew R. Dalby, PhD

Academic Editor

PLOS ONE

Journal Requirements:

2. Please list the name and version of any software package used for statistical analysis, alongside any relevant references. For more information on PLOS ONE's expectations for statistical reporting, please see https://journals.plos.org/plosone/s/submission-guidelines.#loc-statistical-reporting.

3. In your Methods section, please provide additional information about the participant recruitment method and the demographic details of your participants. Please ensure you have provided sufficient details to replicate the analyses such as:

- the recruitment date range (month and year)

- a description of any inclusion/exclusion criteria that were applied to participant recruitment

- a table of relevant demographic details

- a statement as to whether your sample can be considered representative of a larger population

- a description of how participants were recruited.

4. Please include additional information regarding the survey or questionnaire used in the study and ensure that you have provided sufficient details that others could replicate the analyses. For instance, if you developed a questionnaire as part of this study and it is not under a copyright more restrictive than CC-BY, please include a copy, in both the original language and English, as Supporting Information.

5. In the Methods, please clarify that participants provided oral consent. Please also state in the Methods:

- Why written consent could not be obtained

- Whether the Institutional Review Board (IRB) approved use of oral consent

- How oral consent was documented

For more information, please see our guidelines for human subjects research: https://journals.plos.org/plosone/s/submission-guidelines#loc-human-subjects-research

Reviewers' comments:

Reviewer's Responses to Questions

**Comments to the Author**

1. Is the manuscript technically sound, and do the data support the conclusions?

Reviewer #1: Partly

2. Has the statistical analysis been performed appropriately and rigorously? 

Reviewer #1: No

3. Have the authors made all data underlying the findings in their manuscript fully available?

Reviewer #1: No

4. Is the manuscript presented in an intelligible fashion and written in standard English?

Reviewer #1: Yes

5. Review Comments to the Author

Reviewer #1: This study aims to describe risk factors for risk factors for risky sexual practices and prevalence of STI in a population of female sex workers in Quito, Ecuador. This is an important and understudied topic but challenging to study due to the social standing and limited resources of this population. The authors should consider a major revision to the manuscript highlighting the major limitations of these data.

Major comments:

Study aims:

The study aims in this manuscript are not well described. Outlining specific aims and describing the variables and statistical analyses required for each aim will make the study much clearer. This appears to be a descriptive or exploratory study. If that is the case, this should be stated explicitly. Furthermore, the rationale for the exposures and outcomes in this study does not seem well supported. For example, why would previous STI be a risk factor for current risky sexual practices? Would the relationship be the other direction, with RSP causing STI? If the authors hypothesize previous STI lessens the risk of current RSP, this should be stated. In fact, justification for all associations tested should be included.

On the other hand, if this is solely a descriptive study, multivariable regression and other inferential statistics may not be appropriate. However, given this is a convenience sample, the authors have to be very cautious in their interpretations of prevalence.

Sampling:

The manuscript states a convenience sample was used and “cases were recruited through contacts with the association of sex workers in Quito.” There are several things that need to be clarified regarding the sampling methods. First, it is unclear why the word “cases” was used here as not everyone, presumably, was a case. Perhaps the authors intended to use the word participants. In addition, it is unclear how these female sex workers were recruited. More detail needs to be added, such as whether this was a snowball sampling method or something similar. Although these data may not exist, it would also be helpful to know how many women were approached and how many declined. Finally, what is meant by association of sex workers? Is this one organization to which registered all sex workers belong? If so, were all sex workers given an opportunity to participate? If not, what was the strategy for engaging this association? Did the researchers send a formal letter or approach the offices in person? In addition, the manuscript states samples were collected as the women attended a medical consultation required to validate their IHC during the last quarter of 2017. Does this mean only women who had a scheduled medical consultation during this time were eligible to participate? Were the samples collected as part of the routine exam or as an additional measure collected during the study? Much more detail is needed in this section.

Measurement of exposure and outcome:

More detail is needed regarding the collection of data on the exposures. Were they all collected via a self-reported questionnaire? Or were things like FSW association membership collected from other records? Why wasn’t previous STI validated with medical records if the women were attending a clinic? Were the medical records unavailable to the researchers? The authors mention in one sentence that they’re exploring variables associated with being a FSW and then elsewhere they discuss RSP as an outcome variable. Moreover, the authors have not presented a theoretical (or otherwise) justification for studying age or leaving the city for sex work as it relates to membership in a FWS organization.

STIs are associated with a great deal of stigma and may not be reported reliably. Self-reported RSP suffers from the same limitations. Social response/social desirability bias and recall bias may be of concern here. The authors mention this briefly in the limitations section of the conclusion but don’t discuss how this may impact their results. Would this be differential or non-differential misclassification? In other words, would those with STI be more or less likely to report RSP? This should be discussed.

Statistical methods:

As stated above, estimating prevalence from a convenience sample is highly problematic, particularly given we don’t know the characteristics of the entire population and whether these women are representative of all sex workers in Quito.

The authors need to provide more detail about how they conducted the regressions and how they constructed Table 1. Why are the n’s so much greater for some variables than the women who participated in the study? What variables were included in the adjusted OR? How were those variables selected? How was missingness of variables handled?

Finally, why were t tests for age only done for FSW membership and RSP? In addition, were homogeneity of variance and normality considered?

Conclusions:

The authors discuss how membership in an FSW may have affected their results as these sex workers are less likely to work in regulated brothels. This information should be in the introduction as it is important. Also, there is no discussion of why previous report of STI is associated with RSP but treatment for STI is not. Finally, the incidence of various STIs in this sample is somewhat low. This is good news. The authors should consider implications for practice/policy with regard to organized/legal sex work in the country as well as similar countries with legal sex work.

Minor comments:

There appear to be some word spacing and formatting issues throughout the manuscript. For example, sometimes there is a space between the citation and sometimes there is not. Also, some words have several spaces between them and others only one.

6. PLOS authors have the option to publish the peer review history of their article (what does this mean?). If published, this will include your full peer review and any attached files.

Reviewer #1: No

---

## [Author Response · Author response to Decision Letter 0]

23 Mar 2021

Academic editor comments 

Comment 1: Please ensure that your manuscript meets PLOS ONE's style requirements, including those for file naming. 

Response 1: We appreciate the Academic editor comments. We have ensured the manuscript meets PLOS ONE´s style requirements. 

Comment 2: Please list the name and version of any software package used for statistical analysis, alongside any relevant references. 

Response 2: We have now added the name and version of the software package on Statistical analysis, page 6

Comment 3: In your Methods section, please provide additional information about the participant recruitment method and the demographic details of your participants. Please ensure you have provided sufficient details to replicate the analyses such as:

- the recruitment date range (month and year)

- a description of any inclusion/exclusion criteria that were applied to participant recruitment

- a table of relevant demographic details

- a statement as to whether your sample can be considered representative of a larger population

- a description of how participants were recruited.

Response 3: We have added additional information (Material and methods, page 5 and Results, pages 7,8)

Comment 4. Please include additional information regarding the survey or questionnaire used in the study and ensure that you have provided sufficient details that others could replicate the analyses. For instance, if you developed a questionnaire as part of this study and it is not under a copyright more restrictive than CC-BY, please include a copy, in both the original language and English, as Supporting Information.

Response 4. We have included the questionary as Supporting information 4. The Data were added as Supporting information 3. 

Comment 5. In the Methods, please clarify that participants provided oral consent. Please also state in the Methods:

- Why written consent could not be obtained

- Whether the Institutional Review Board (IRB) approved use of oral consent

- How oral consent was documented

 Response 5: In the methods, page 5, we explain that “Written and oral informed consent was obtained from all participants”

Comments from reviewer #1

This study aims to describe risk factors for risk factors for risky sexual practices and prevalence of STI in a population of female sex workers in Quito, Ecuador. This is an important and understudied topic but challenging to study due to the social standing and limited resources of this population. The authors should consider a major revision to the manuscript highlighting the major limitations of these data.

Major comments:

Study aims:

Reviewer´s comment 1: The study aims in this manuscript are not well described. Outlining specific aims and describing the variables and statistical analyses required for each aim will make the study much clearer.

Response 1: We appreciate the reviewer´s positive comments. We have outlined specific aims on Introduction, page 4

Reviewer´s comment 2: This appears to be a descriptive or exploratory study. If that is the case, this should be stated explicitly. Furthermore, the rationale for the exposures and outcomes in this study does not seem well supported. For example, why would previous STI be a risk factor for current risky sexual practices? Would the relationship be the other direction, with RSP causing STI? If the authors hypothesize previous STI lessens the risk of current RSP, this should be stated. In fact, justification for all associations tested should be included.

Response 2: We have added information about the kind of study on Methods, page 5; information about associations on Introduction, page 3-4 and on Discussion, page 11 (information about diagnosis of STI could be an indirect indicator of RSP)

Reviewer´s comment 3: On the other hand, if this is solely a descriptive study, multivariable regression and other inferential statistics may not be appropriate. However, given this is a convenience sample, the authors have to be very cautious in their interpretations of prevalence.

Response 3: It is a multivariate descriptive that seeks association, does not seek causality. It was not the objective of the study to estimate rates at a national level, but rather to estimate proportion of STI in this population of female sex workers. 

Sampling:

Reviewer´s comment 4: The manuscript states a convenience sample was used and “cases were recruited through contacts with the association of sex workers in Quito.” There are several things that need to be clarified regarding the sampling methods. First, it is unclear why the word “cases” was used here as not everyone, presumably, was a case. Perhaps the authors intended to use the word participants.

Response 4: We have added a statement that clarified convenience sampling on Methods, page 5. We have corrected de word “cases” by participants.

Reviewer´s comment 5: In addition, it is unclear how these female sex workers were recruited. More detail needs to be added, such as whether this was a snowball sampling method or something similar. Although these data may not exist, it would also be helpful to know how many women were approached and how many declined. 

Response 5: We have added a statement that clarified convenience sampling on Methods, page 5 and we mentioned on the Results, page 7 “Of 251 FSWs invited, 249 consented to participate”

Reviewer´s comment 6: Finally, what is meant by association of sex workers? Is this one organization to which registered all sex workers belong? If so, were all sex workers given an opportunity to participate? If not, what was the strategy for engaging this association? Did the researchers send a formal letter or approach the offices in person?

Response 6: We explained what is a FSWs association on Introduction, page 5, added a statement on Methods, page 5 and on Discussion, page 11. 

Reviewer´s comment 7: In addition, the manuscript states samples were collected as the women attended a medical consultation required to validate their IHC during the last quarter of 2017. Does this mean only women who had a scheduled medical consultation during this time were eligible to participate? 

Response 7: The women who were eligible to participate were who attended to medical consultation and the women from FSW´s association. 

Reviewer´s comment 8: Were the samples collected as part of the routine exam or as an additional measure collected during the study? Much more detail is needed in this section.

Response 8: An extra sample was collected during the routine exam. We added this information on Clinical samples collection and DNA preparation, page 6 

Measurement of exposure and outcome:

More detail is needed regarding the collection of data on the exposures. 

Reviewer´s comment 9: Were they all collected via a self-reported questionnaire? Or were things like FSW association membership collected from other records? Why wasn’t previous STI validated with medical records if the women were attending a clinic? Were the medical records unavailable to the researchers? 

Response 9: On Methods, page 5. We wrote that “Participants were interviewed by a local study investigator (LMLLA) using a questionnaire”. We did not have access to medical records. 

Reviewer´s comment 10: The authors mention in one sentence that they’re exploring variables associated with being a FSW and then elsewhere they discuss RSP as an outcome variable. Moreover, the authors have not presented a theoretical (or otherwise) justification for studying age or leaving the city for sex work as it relates to membership in a FWS organization.

Response 10: The outcome variable is RSP. We have clarified that there are factors related with RSP on Introduction, page 4 and Discussion, page 12

Reviewer´s comment 11: STIs are associated with a great deal of stigma and may not be reported reliably. Self-reported RSP suffers from the same limitations. Social response/social desirability bias and recall bias may be of concern here. The authors mention this briefly in the limitations section of the conclusion but don’t discuss how this may impact their results. Would this be differential or non-differential misclassification? In other words, would those with STI be more or less likely to report RSP? This should be discussed.

Response 11: We agree that the self- report about RSP could result in underreporting for that we suggest that ask regarding some aspects of STI is another way to get more reliable information about sexual behaviour. We have provided a statement on Discussion, page 12. 

Reviewer´s comment 12:

Statistical methods:

As stated above, estimating prevalence from a convenience sample is highly problematic, particularly given we do not know the characteristics of the entire population and whether these women are representative of all sex workers in Quito.

Response 12. We have clarified that for the sampling method the results cannot be extrapolated to the entire FSW population. Discussion, page 13.

Reviewer´s comment 13: The authors need to provide more detail about how they conducted the regressions and how they constructed Table 1. Why are the n’s so much greater for some variables than the women who participated in the study? What variables were included in the adjusted OR? How were those variables selected? How was missingness of variables handled?

Response 13: There was a mistake in the Table’s format and the numbers in columns were joined. Correction was made. 

This is a descriptive study; the model was not introduced step by step but with the entire block of variables. The selected variables correspond to a previous working hypothesis. Our variable of interest is RSP. We added the regression model as Supporting information. 

We could not get the PCR information about four samples hence these were no considered in the STI’s proportion calculating. We have indicated this on Methods-PCR screening for STI, page 6 

Reviewer´s comment 14: Finally, why were t tests for age only done for FSW membership and RSP? In addition, were homogeneity of variance and normality considered?

Response 14: The t test for age was performed for the other variables, but it was not significant. Although, the variable is not normal, the means are calculated by the Central Limit Theorem as normal. Using Levene's test, the homogeneity of variances was considered. Once the homogeneity of variances has been established, the t test is robust and detects differences when there are any. 

Conclusions:

Reviewer´s comment 15: The authors discuss how membership in an FSW may have affected their results as these sex workers are less likely to work in regulated brothels. This information should be in the introduction as it is important.

Response 15: We have added a statement about the place of work of members of association. Introduction, page 4 

Reviewer´s comment 16: Also, there is no discussion of why previous report of STI is associated with RSP but treatment for STI is not. 

Response 16: The information about STI and RSP could be under-reported by recall bias leading in inaccurate response. We have added information on Discussion, page 12

Reviewer´s comment 17: Finally, the incidence of various STIs in this sample is somewhat low. This is good news. The authors should consider implications for practice/policy regarding organized/legal sex work in the country as well as similar countries with legal sex work.

Response 17: We have considered this on Discussion, page 13.

Minor comments:

Reviewer´s comment 18: There appear to be some word spacing and formatting issues throughout the manuscript. For example, sometimes there is a space between the citation and sometimes there is not. Also, some words have several spaces between them and others only one.

Response 18: We have revised that.

---

## [Editor Report · Decision Letter 1]

31 Mar 2021

Sexually transmitted infections and factors associated with risky sexual practices among female sex workers: a cross sectional study in a large Andean city

PONE-D-20-32212R1

Dear Dr. Llangarí,

We’re pleased to inform you that your manuscript has been judged scientifically suitable for publication and will be formally accepted for publication once it meets all outstanding technical requirements.

Kind regards,

Andrew R. Dalby, PhD

Academic Editor

PLOS ONE
---

## [Editor Report · Acceptance letter]

27 Apr 2021

PONE-D-20-32212R1 

Sexually transmitted infections and factors associated with risky sexual practices among female sex workers: a cross sectional study in a large Andean city  

Dear Dr. Llangarí-Arizo:

I'm pleased to inform you that your manuscript has been deemed suitable for publication in PLOS ONE. Congratulations! Your manuscript is now with our production department. 

Kind regards, 

on behalf of

Dr. Andrew R. Dalby 

Academic Editor

PLOS ONE